# Paper-Based Electrodes Conjugated with Tungsten Disulfide Nanostructure and Aptamer for Impedimetric Detection of *Listeria monocytogenes*

**DOI:** 10.3390/bios12020088

**Published:** 2022-01-31

**Authors:** Annu Mishra, Roberto Pilloton, Swati Jain, Souradeep Roy, Manika Khanuja, Ashish Mathur, Jagriti Narang

**Affiliations:** 1Amity Institute of Nanotechnology, Amity University, Noida 201313, India; annum407@gmail.com (A.M.); swatijain.iitd@gmail.com (S.J.); 2CNR-IC, Area dellaRicerca di RM1, Via Salaria km 29.3, Monterotondo, I-00015 Rome, Italy; 3Centre for Interdisciplinary Research and Innovation (CIDRI), University of Petroleum and Energy Studies, Dehradun 248007, India; roysouradeep04@gmail.com (S.R.); nanoashish@gmail.com (A.M.); 4Centre for Nanoscience and Nanotechnology, Jamia Millia Islamia, New Delhi 110025, India; manikakhanuja@gmail.com; 5Department of Biotechnology, Jamia Hamdard, New Delhi 110062, India; jags_biotech@yahoo.co.in

**Keywords:** ePAD, *Listeria monocytogenes*, WS_2_ nanoparticles, paper-based electrode, electrochemical aptasensor, point-of-care

## Abstract

In this study, we report on a novel aptasensor based on an electrochemical paper-based analytical device (ePAD) that employs a tungsten disulfide (WS_2_)/aptamer hybrid for the detection of *Listeria monocytogenes.* Listeria is a well-known causative pathogen for foodborne diseases. The proposed aptasensor signifies many lucrative features which include simple, cost-effective, reliable, and disposable. Furthermore, the use of an aptamer added more advantageous features in the biosensor. The morphological, optical, elemental composition, and phase properties of the synthesized tungsten disulfide (WS_2_) nanostructures were characterized by field-emission scanning electron microscopy (FESEM), RAMAN spectroscopy, photoluminescence (PL), and X-ray diffraction (XRD), while electrochemical impedance spectroscopy was performed to corroborate the immobilization of aptamer and to assess the *L. monocytogenes* sensing performance. The limit of detection (LoD) and limit of quantification (LoQ) of the aptasensor was found to be 10 and 4.5 CFU/mL, respectively, within a linear range of 10^1^–10^8^ CFU/mL. The proposed sensor was found to be selective solely towards *Listeria monocytogenes* in the presence of various bacterial species such as *Escherichia coli* and *Bacillus subtilis*. Validation of the aptasensor operation was also evaluated in real samples by spiking them with fixed concentrations (10^1^, 10^3^, and 10^5^) of *Listeria monocytogenes*, thereby, paving the way for its potential in a point-of-care scenario.

## 1. Introduction

Foodborne pathogens have proven to be a severe threat to the food industries that deal with important public health crises across the world [1]. According to the World Health Organization (WHO), foodborne diseases affect approximately one-third of the population in developed countries annually, while this ratio is even higher in developing nations [2]. Bacteria such as *Listeria monocytogenes,* a harmful organism (e.g., human listeriosis), is a considerable concern for pregnant women, infants, and adults who are suffering from a weak immune system [3]. *L. monocytogenes* is associated with the consumption of raw and undercooked meat, poultry food, unpasteurized milk, cheese, and other dairy products. Furthermore, reports have suggested that death rates due to *L. monocytogenes* surged between 2001 and 2009 [4,5]. Therefore, the consequences associated with this bacterium are the foremost concern in food industry treatments.

Detection of pathogenic bacteria can be done through conventional techniques that involve a culture-based approach or a combination of a culture-based approach with numerous biochemical assays [6]. In spite of being highly precise and possessing the ability to distinguish between live and dead cells, such assays are bulky, non-specific, less sensitive, time-consuming, lengthy experimental procedures, lack quantitative results, and are unfeasible for point-of-care applications [7]. Due to these limitations, there is an extensive requirement to develop a suitable approach that can address the wide gaps in the existing detection methods.

Electrochemical biosensors have been incorporated as an alternative approach to overcome the drawbacks of traditional methods for foodborne pathogen monitoring, with the focus on being faster, cheaper, and real-time detection [8]. In the present era, an electrochemical biosensor matrix plays an essential role in improving a highly selective, reliable, sensitive and cost-effective analysis for clinical diagnosis [9]. Metallic nanostructures can be incorporated as electrode surface modifiers because they offer high electric conductivity due to quantum confinement effects [10]. Such a property is desirable for enhancing interfacial electron transfer kinetics.

In the present approach, we emphasize the fabrication of a label-free impedimetric aptasensor to detect *Listeria monocytogenes*. Construction of an electrochemical aptasensor involves using a paper-based device with advantageous features such as a low volume of analytes, mass-producible, lightweight, flexible, affordability, disposable, and rapidity, along with the feasibility of liquid flow without any external perturbation. The abovementioned advantages of electrochemical paper analytical devices (ePADs) make them a cheap platform for use in a resource-limited environment [11]. Specifically, the working electrode of anePAD was modified with tungsten disulfide (WS_2_) due to its biocompatibility and higher electron transfer to enhance the electrochemical signals. Furthermore, the aptamer modified with amine groups was immobilized on the surface-modified working electrode of the ePAD and an electrochemical analysis was performed to confirm the respective hybridization. In our current study, the aptamer was selected based on the work done by Sidhu et al. [11] and was exploited to fabricate a biosensor [12].

In this research, we report on the analytical performance of a WS_2_/aptamer hybrid-based aptasensor towards the selective detection of *Listeria monocytogenes* in an experimentally synthesized electrolyte as well as spiked dairy products, thereby, validating the developed sensor as a potential candidate for specific and sensitive bacterial detection.

## 2. Materials and Methods

### 2.1. Reagents

Sodium tungstate (Na_2_WO_4_), thiourea (CH_4_N_2_S), and methylene blue (MB) were purchased from Fisher Scientific, Pvt. Ltd., New Delhi, India. Hydroxylamine hydrochloride (HONH_2_·HCl), polyethylene glycol (C_2n_H_4n+2_O_n+1_), potassium chloride (KCl), sodium monobasic (NaH_2_PO_4_), and dibasic salts (Na_2_HPO_4_) were purchased from Central Drug House (CDH), Pvt. Ltd., New Delhi, India. NaH_2_PO_4_ and Na_2_HPO_4_ were used in the preparation of phosphate buffer saline (PBS, 0.1 M and pH 7.4). Tris HCl-EDTA (TE buffer) was used to prepare the aliquot of the oligonucleotides and, afterwards, stored at 4 °C. All the chemicals used were prepared in double distilled water (DDW).

Aliquots of bacterial strains used, in this study, were acquired from the Culture Collection (*Listeria monocytogenes* strains), Microbial Type Culture Collection, and Gene Bank CSIR-IMTECH, Chandigarh (1143-*L. monocytogenes*). *E. coli* and *B. subtilis* were cultured in sterilized conditions at the Amity University, Amity Institute of Biotechnology, Noida.

The amine-modified sequence of *L. monocytogenes* was synthesized by GCC Biotech (Pvt. Ltd., Delhi, India). The 47-mer aptamer sequence of Listeria was obtained from Sidhu, R. et al. [11], i.e., -5′-NH_2_-ATC CAT GGG GCG GAGATG AGG GGG AGG AGG GCG GGT ACC CGG TTGAT-3′.

### 2.2. Apparatus

Electrochemical impedance spectroscopy was performed using a Wayne Kerr 6500-B Precision Impedance Analyzer for sensing of *L. monocytogenes* on WS_2_/aptamer modified electrodes. The ePAD involved a two-electrode system that was fabricated by screen printing conductive carbon ink on 210 GSM sheets. The working region of the ePAD was incorporated with WS_2_ nanostructures (WS_2_NS) as a biocompatible and enhanced electron conductivity matrix to detect *L. monocytogenes*.

The morphology of the synthesized WS_2_ nanostructure was characterized by FE-SEM (Zeiss 18448), while the molecular fingerprints of the WS_2_ nanostructures were analyzed by RAMAN spectroscopy (Shimadzu 8700), in the region of 500–4500 cm^−1^. The X-ray diffraction patterns (RigakuXRD, Smart Lab) were recorded in the range of 10° to 80° using CuKα radiation (40 kV, 30 mA) to investigate the crystal structure of WS_2_. The photoluminescence (PL) spectroscopy was performed using a RF5301-PC SHIMAZZU spectrofluorometer to analyze the defect structure of the synthesized nanomaterial. The PL spectra were recorded at the excitation wavelength of 650 nm with 300 to 800 nm of scanning through a 1.0 nm step length.

### 2.3. Bacteria Culture

The cultivation of pathogenic strains was performed in Luria broth (LB) medium at 37 °C by stirring for 16 h at 170 rpm. Bacterial cultures were centrifuged at 5800 r/min for 10 min (25 °C) and washed thrice by PBS (0.1 M, pH 7.4). The resulting precipitate was mixed in 15 mL of PBS as the initial stock solution of *L. monocytogenes*. The bacteria concentrations were prepared by serial dilution in PBS, while the corresponding absorbance was measured at 600 nm. Then, the original bacterial aliquot was further diluted to prepare concentrations from 10^1^ to 10^8^ CFU/mL

### 2.4. Synthesis of Tungsten Disulfide (WS_2_) Nanoflakes

Sodium tungstate (4.74 g), thiourea (4.5 g), hydroxylamine hydrochloride (2.06 g), and PEG (0.54 g) were added to 90 mL of deionized (DI) water under constant stirring in a beaker for 1h.The prepared solutions were transferred to a 150 mL Teflon-lined autoclave and kept at 180 °C for 24 h. After 24 h, the autoclave was allowed to cool down at room temperature. The obtained sample was subjected to vacuum filtration using Whatman filter paper. The filtrate was washed several times with ethanol and DI water, followed by drying at 40 °C for 4h. The obtained sample was ground to form a fine powder.

### 2.5. Development of Paper-Based Sensor Drop Cast with Tungsten Disulfide Nanoflakes (WS_2_NFs)

The electrochemical paper-based micro-analytical device (ePAD)was fabricated via screen printing technology using carbon conductive ink, as reported in the literature [13,14]. After drying the ink for 20 min, a wax coating was applied around the electrodes to create a hydrophobic barrier and to define the working area over the ePAD. Further, the working electrodes were modified with WS_2_nanostructure (10 μL) drop cast on the ePAD and were dried at 25 °C, followed by immobilization of 5μL aptamer (20 µM), and allowed to stabilize for overnight at 4 °C.

### 2.6. Fabrication of Listeria Aptamer ssDNA/WS_2_NF/ePAD

In order to fabricate the Listeria aptamer single-stranded DNA/WS_2_NF/ePAD modified electrode, various concentrations (20, 40, 60, 80, and 100 μM) of specific aptamer sequences of *L. monocytogenes* were optimized. The dilution made different concentrations of the initial 100 μM ssDNA solution with Tris-EDTA (TE) buffer.

The ssDNA aptamer (5′-NH_2_-ATC CAT GGG GCG GAGATG AGG GGG AGG AGG GCG GGT ACC CGG TTGAT-3′) of *L. monocytogenes* (5 μL) was immobilized on the WS_2_NF-coatedePAD, at 4 °C for 1 h. The electrochemical impedance spectroscopy studies were recorded using 0.1 M KCl electrolyte with 0.1 mM MB. The performance of the aptasensor was analyzed by exposing it to solutions consisting of various concentrations of the bacteria. The overall process is summarized in Figure 1.

### 2.7. Bacteria Detection

First, the sensor was incubated using Listeria concentration (10^8^ CFU/mL) at 37 °C until the bacteria bound completely (30 min) with corresponding aminated aptamer. Then, the EIS analysis was performed in 0.1 M KCl containing 0.1 mM MB.

Control bacteria such as *E. coli* and *B. subtilis* were employed for the specificity and standard experiments. The fabrication was the same for L. monocytogenes (target) detection as procedure followed in aptamer fabrication. Then, each bacterium, apart from Listeria to detect specificity and along with bacteria to detect selectivity, was used for the electrochemical detection at the same concentration (10^8^ CFU/mL).

### 2.8. Repeatability and Storage Stability

The repeatability of the aptasensor was checked three times on the same day (intra-day) and duplicate measurements were performed under the same conditions after three days (inter-day) and employed for the detection of the same target bacteria concentration. The fabricated ePAD incorporated with nanomaterial was checked for 20 days at regular intervals of five days, after storage at 4 °C for the analysis of long-term stability.

### 2.9. Application of ssDNA/WS_2_NS/ePAD in Food Samples

The developed aptasensor was also exposed to food sources of *L. monocytogenes,* such as unpasteurized milk and soft cheese made from unpasteurized milk. In the current process, dilutions of dairy products were done with 5 mL PBS (pH 7.4). The prepared samples were further centrifuged for 10 min at 10,000 rpm. Bacteria were spiked in dairy samples to assess the performance of the aptasensor. Afterward, the EIS study was performed to detect *L. monocytogenes* in the presence of *E. coli* and *B. subtilis* in the respective dairy sample on the aptamer-modified ePAD electrode surface.

## 3. Results and Discussion

### 3.1. Surface Characterization of WS_2_ Nanostructures

The morphological and structural properties of synthesized WS_2_ were performed by FESEM, XRD, RAMAN, and PL. The SEM micrograph shown in Figure 1a indicates the randomly distributed, densely packed flaky morphology of WS_2_ nanostructures. The length and width of the flakes were measured to be ~400 and 200 nm, respectively. Figure 1b shows the Raman spectrum of as-synthesized WS_2_ nanostructures. The Raman spectrum depicted E_2g_^1^ and A_1g_ which are two active modes. The E_2g_^1^ mode arises from in-phase vibrations of W atoms in the opposite direction with respect to S atoms, while the A_1g_ mode is assigned to the S atoms which are moving in-phase and in out-of-plane directions. We observed the E_2g_^1^ mode at 351 cm^−1^ and the A_1g_ mode at 420 cm^−1^; the difference between these two modes, i.e., Δω was 69 cm^−1^, which confirmed the formation of WS_2_ [13].

The diffraction peaks of WS_2_with their respective (hkl) planes were obtained at 15.09° (002), 23.4° (112), 28.2° (004), 29.2° (100), 30.5° (101), 35.3° (102), 38.64° (103), 46.4° (006), 50.82° (105), 53.22° (106), 59.55° (110), 65.51° (114), and 71.22° (203), which confirmed the formation of WS_2_ with hexagonal structure (JCPDS No.84-1398, 08-0237) (Figure 1c). A sharp peak (002) indicated the highly crystalline nature of synthesized WS_2_ [13,15], while the small one at 23.41° was due to the organic moiety of polyethylene glycol (PEG), which was used as a surfactant.

Figure 1d shows the PL spectra of WS_2_ nanostructures for the evaluation of band gap and structural defects. The peak around 650 nm is attributed to the direct band-edge transition from conduction band minima to valence band maxima and the corresponding bandgap is 1.9 eV [16].

### 3.2. Electrochemical Characterization of ePADs

An EIS analysis was performed to characterize the stepwise assembly of the ePADs obtained after modification with WS_2_NS, aptamer ssDNA, or bacteria in 0.1 M KCl with MB(0.1 mM).The Nyquist spectra (shown in Figure 2a–c) indicate a semicircular feature with a halfarc arising at the high-frequency region at the left side of each spectrum. The high-frequency region essentially gives the bulk electrode/electrolyte signature, in which case, the halfarc indicates a low capacitance (C), as corroborated by high-capacitive impedance Z″ (since Z″ α C^−1^) [17].

It is possible to speculate that the initial low capacitance could be a signature of the screen-printed paper electrode, whereby, the poor dielectric constant (C α ε) of the base cellulose sheet appears to contribute towards poor charge storage ability at higher frequencies [18]. Moreover, the Nyquist spectrum seems to be shifted to the right along the Z’ axis which indicates the presence of bulk resistance of base electrode and electrolyte, the latter consisting of 0.1 mM methylene blue probe in 0.1 M KCl [19]. The spectrum can be modeled by a series of resistance R_s_ (see Randel’s circuit of Figure 3a). Furthermore, the mid-frequency region gives a measure of heterogeneous electron transfer kinetics, which is the redox process of methylene blue probe (0.1 mM in 0.1 M KCl). The electrode/electrolyte interface can be modeled using a parallel combination of resistor (indicating charge transfer resistance, R_ct_) and double-layer capacitance (C_dl_), as corroborated by the presence of a typical semicircular feature in the mid-frequency spectra. This situation denotes a turnover of electron traversal from capacitive to resistive paths since the latter is effectively dominant at low frequencies. Therefore, the parallel arrangement ensures two possible pathways for electron flow, which occur via the capacitive approach at higher frequencies (Z″ α f^−1^) and through the resistor at low frequencies [20]. Finally, the low-frequency region indicates the occurrence of diffusion-mediated transport of methylene blue probe towards the electrode surface for undergoing its signature redox process, thereby, generating electrons at the electrode/electrolyte interface.

Now, Figure 2a depicts the Nyquist plot at different stages of electrode fabrication which shows a huge R_ct_ of ~101.91 kΩ for the bare ePAD but a similar semi-circle with lower R_ct_ value (~46.441 kΩ) for the WS_2_NS/ePAD. After immobilization of aptamers onto WS_2_NS, the R_ct_ value increased due to the non-conductive nature of aptamers, along with the repulsion between negatively charged sequence and ions present in the electrolyte. After incubation with bacteria, the resistance increased due to the non-conductive nature of bacteria bonded to the aptamer. The EIS results endorse the successful fabrication of the aptasensor.

Figure 2b shows the impedance response of modified electrodes against different aptamer concentrations. During this experiment, it was observed that the R_ct_ was the least at 20 µM concentration and increased further with increasing concentration of aptamer. The intensification in R_ct_ value with increasing aptamer concentration ascribed to the deposition of a more insulating layer of DNA aptamer onto the working surface of ePAD. Therefore, further studies were performed at a lower aptamer concentration (20 µM).

A time-dependent study for the sensor fabrication of ePAD was also performed. The effects are shown in Figure 2c at different immobilization times (5, 10, 15, 20, 25, and 30 min) by incubating the bacteria/aptamer/WS_2_NS/ePAD. The impedimetric response obtained after incubating for 25 and 30 min differed insignificantly due to the enhanced binding between the aptamer and bacteria. Therefore, the optimum time was chosen to be 25 min, as sufficient interaction of bacteria with the aptamer.

### 3.3. Electrochemical Impedance Spectroscopic Analysis of Listeria monocytogenes

Different concentrations (10^1^, 10^2^, 10^3^, 10^4^, 10^5^, 10^6^, 10^7^, and 10^8^ CFU/mL) of bacteria were incubated on the modified ePAD and the results are shown in Figure 3a. It is interesting to note that the Nyquist profile at various *L. monocytogenes* target concentrations retain the features shown in Figure 2a–c. Specifically, the low capacitance due to the poor dielectric constant of paper substrate and bulk electrode/electrolyte resistance at high frequencies, the mid-frequency interface kinetics, and low frequency methylene blue diffusion appear to be preserved in the concentration scan, which essentially indicates the stability of the fabricated aptamer/WS_2_ bio-nano hybrid-coated ePAD.

During the apta-recognition process, as the bacterial concentration is increased, more of the bacteria becomes bonded to the probe-aptamer sites and passivates the electrode surface due to the insulating nature of bacteria [21], and the interfacial charge transfer is drastically reduced. This is indicated in Figure 3a by an increase in R_ct_ from ~58 to 96 kΩ upon elevating the *L. monocytogenes* concentration from 10^1^ to 10^8^ CFU/mL. This situation has immediate consequences on the double layer capacitance C_dl_. It can also be observed, in Figure 3a, that C_dl_ appears to decrease with an increase in bacterial concentration, which can be attributed to poor double layer charging at higher concentrations facilitated by reduced electron transfer (high R_ct_) at the electrode/electrolyte interface. The Randel’s circuit, shown in the inset in Figure 3a, was employed to obtain approximate estimates of the interfacial parameters involved in the *L. monocytogenes* sensing scenario by curve fitting of the Nyquist spectra of Figure 3a using Zview software. The latter is highlighted in Figure 3b, while the values of various interfacial parameters are listed in Table 1.

It can be observed from the above table that an increase in R_ct_ from 58 to 96 kΩ has a drastic impact on C_dl_, as indicated in by the three-fold increase from 2 µF to 5 nF, as the *L. monocytogenes* concentration is scaled up from 10^1^ to 10^8^ CFU/mL. As mentioned above, this is primarily due to poor double layer charging, since fewer electrons traverse the interface region due to the increased binding of the target and probe aptamer on the electrode surface. While the high-frequency bulk resistance remains constant at 5 kΩ, the diffusion impedance W_s_ increases from 0.1 to 0.6 kΩ due to steric hindrance between methylene blue and target aptamers at higher concentrations.

The calibration plot, obtained at a frequency of 120.45 kHz (Figure 3c) showed that Rct increased linearly as the bacteria concentration was increased from 10^1^ to 10^8^ CFU/mL. These values also corresponded to the diameter of the semi-circle. The calibration plot (Figure 3b) verified the direct proportionality of R_ct_ and bacteria concentration, as it increased along with the concentration range from 10^1^ to 10^8^ CFU/mL. The limit of detection (LOD) and limit of quantification (LOQ) were graphically determined by the Meier and Zund method [22], i.e., 10.0 CFU/mL and 4.5 CFU/mL, respectively.

### 3.4. Selectivity of the Aptasensor

In order to validate the selectivity of the developed sensor, a sensing analysis was performed with various analytes. An interfering study with two different bacterial species, i.e., *E. coli* and *B. subtilis,* was conducted to validate the selectivity of the developed sensor. Figure 4 shows that each bacterium generates its own unique charge transfer response (R_ct_) with significant and specific deviations. Furthermore, a fixed concentration of *E. coli* and *B. subtilis* (10^8^ CFU/mL) was individually mixed with *L.*
*monocytogenes* (10^8^ CFU/mL) and the R_ct_ values of these mixtures were recorded at a calibration frequency of 120.45 kHz. As shown in Figure 4, the R_ct_ value for *L. monocytogenes* alone and the mixtures with *E. coli* and *B. subtilis* were ~95.1 ± 3.8, 97.2 ± 3.5, and 93.8 ± 3.9 kΩ, respectively. These results indicate that irrespective of the presence of different bacteria in the sample, the developed sensor can accurately detect the presence of *L. monocytogenes,* which is attributed to its selective binding with the aptamer sequence. Therefore, from the result, it is concluded that the aptasensor responds selectively to *L. monocytogenes*.

### 3.5. Reproducibility and Storage Stability

Reproducibility of the ePAD was also studied and the coefficients of variation of four electrodes were tested and determined to be 52.503 kΩ ±1.18% (mean of R_ct_ ± COV, within the batch) and 58.365 kΩ ± 5.54% (mean of R_ct_ ± COV, between the batch). The developed sensor was also tested for its stability and shelf-life. The impedance response of the sensor was recorded at a concentration of 10^1^ CFU/mL for 20 days (Figure 5) at a regular interval of 5 days. The observed average value of impedance was found to be ~64.4 ± 6.4 kΩ; and further, it indicated that the impedance changed insignificantly for 15 days, suggesting that the sensor gives stable impedance response over a period of 15 days.

### 3.6. Application of the Aptasensor in Real Sample Analysis

The validation of our aptasensor was analyzed by spiking milk and cheese with bacteria concentrations of 10^1^, 10^3^, and 10^5^ CFU/mL. The bar graph below (Figure 6) indicates that interfacial charge transfer resistance in spiked milk and cheese samples differs insignificantly as compared with that observed for the target. The minor increment in impedance observed for spiked cheese and milk for the target may be attributed to the former being insulating in nature, contrary to what is observed for the latter, which is suspended in a conducting electrolyte (MB/KCl). Furthermore, Table 2 shows the percentage recovery of bacteria in cheese and milk. Such high recovery proves the validation of the developed aptasensor.

## 4. Conclusions

In the present work, we demonstrated the feasibility of a paper-based platform utilizing WS_2_ for the detection of foodborne pathogens. An electrochemical sensor based on a WS_2_/aptamer hybrid was successfully fabricated to detect Listeria. Impedance spectroscopy was performed using a two-electrode system. The use of the paper-based platform reduced the cost of the sensor fabrication and the volume of the analyte required. The linear range for the detection of Listeria was 10^1^–10^8^ CFU/mL with a detection limit of 10.0 CFU/mL. The constructed aptasensor demonstrated high selectivity, reproducibility, and sensitivity toward *L. monocytogenes*. Due to the impressive detection limits and linear range, the developed sensor can be implemented in real-time field applications.

## Data Availability

We have not reported any data where supporting reported results can be found.

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
