# Peer review of "Paper-Based Electrodes Conjugated with Tungsten Disulfide Nanostructure and Aptamer for Impedimetric Detection of Listeria monocytogenes"

_biosensors, 2022, doi:10.3390/bios12020088_

Round 1

Reviewer 1 Report

Mishra et al. report on the “Paper-based electrodes conjugated with tungsten disulfide nanoparticles and aptamer for impedimetric detection of Listeria 3 monocytogenes.” The content of the work is interesting, but the manuscript cannot be published in the present form due to the following issues:

  1. The authors have claimed the formation of WS2 nanoparticles in the title. Do they really synthesize WS2 nanoparticles if so then what is its size? The authors have requested sot provide some evidence of the formation of WS2 nanoparticles? If not then the title of the manuscript should be changed accordingly.
  2. The EDS is important to understand the composition of the structures shown in Figure 1 (a) as FESEM. It is mandatory to include?
  3. Rather than FTIR which is shown in Figure 1 (b), Raman is required for the understanding of the formation of WS2?
  4. The peak which is shown at 650 nm in Figure 1 (d) should be magnified for proper understanding.
  5. There is no equivalent circuit diagram that is used by the author to fit the curves of Figure 2 (a-c). Also, the fitting curve is mandatory?
  6. Explanation of the behavior of the curves at high frequency and low-frequency range for all the Nyquist plots (i.e. Figure 2 (a-c), Figure 3 (a) is missing. It's mandatory to include.
  7. The table is required that includes the values of Rs, Rct, Cdl, and Ws for Figure 3 (a)

Author Response

attached file

Reviewer 2 Report

Title: Paper-based electrodes conjugated with tungsten disulfide nanoparticles and aptamer for impedimetric detection of Listeria monocytogenes, this is written and organized well, suitable experiments and characterizations have been done, so it can be published in B after minor revision.

  1. About, “Synthesis of Tungsten Disulphide” did you use ultrasonic dispersion before using WS2, As I can see in fig. 1a, the nanosheets look aggregated, and using ultrasonic is necessary.
  2. Real Sample Analysis, please indicate, how many bacteria is existing in the real sample, naturally? Do you have any information about the number of bacteria in a polluted /rot cheese? Please add the standard levels of bacteria in chees and milk industry? The presented sensor can help and improve the monitoring process in the food industry?
  3. About the LOD of the sensor, could you provide a summary that shows the developed sensor has higher capabilities compared to the reported sensors for a similar target?
  4. please add these references to improve your work

---- Wachiralurpan, Sirirat, Isaratat Phung-On, Narong Chanlek, Supatra Areekit, Kosum Chansiri, and Peter A. Lieberzeit. "In-Situ Monitoring of Real-Time Loop-Mediated Isothermal Amplification with QCM: Detecting Listeria monocytogenes." Biosensors 11, no. 9 (2021): 308.

----- Busa, Lori Shayne Alamo, Saeed Mohammadi, Masatoshi Maeki, Akihiko Ishida, Hirofumi Tani, and Manabu Tokeshi. "Advances in microfluidic paper-based analytical devices for food and water analysis." Micromachines 7, no. 5 (2016): 86.

----- Kumar, Saurabh, Chandra Mouli Pandey, Amir Hatamie, Abdolreza Simchi, Magnus Willander, and Bansi D. Malhotra. "Nanomaterial‐Modified Conducting Paper: Fabrication, Properties, and Emerging Biomedical Applications." Global Challenges 3, no. 12 (2019): 1900041.

------ Oliveira, Daniela A., Suleiman Althawab, Eric S. McLamore, and Carmen L. Gomes. "One-Step Fabrication of Stimuli-Responsive Chitosan-Platinum Brushes for Listeria monocytogenes Detection." Biosensors 11, no. 12 (2021): 511.

Author Response

attached file

Round 2

Reviewer 1 Report

The revised manuscript is acceptable for publication